# Cytomegalovirus and Glioblastoma: A Review of the Biological Associations and Therapeutic Strategies

**DOI:** 10.3390/jcm11175221

**Published:** 2022-09-04

**Authors:** Tianrui Yang, Delin Liu, Shiyuan Fang, Wenbin Ma, Yu Wang

**Affiliations:** 1Department of Neurosurgery, Peking Union Medical College Hospital, Chinese Academy of Medical Sciences and Peking Union Medical College, Beijing 100730, China; 2Department of Neurology, Peking Union Medical College Hospital, Chinese Academy of Medical Sciences and Peking Union Medical College, Beijing 100730, China

**Keywords:** cytomegalovirus, glioblastoma, carcinogenesis, therapeutics

## Abstract

Glioblastoma is the most common and aggressive malignancy in the adult central nervous system. Cytomegalovirus (CMV) plays a crucial role in the pathogenesis and treatment of glioblastoma. We reviewed the epidemiology of CMV in gliomas, the mechanism of CMV-related carcinogenesis, and its therapeutic strategies, offering further clinical practice insights. To date, the CMV infection rate in glioblastoma is controversial, while mounting studies have suggested a high infection rate. The carcinogenesis mechanism of CMV has been investigated in relation to various aspects, including oncomodulation, oncogenic features, tumor microenvironment regulation, epithelial–mesenchymal transition, and overall immune system regulation. In clinical practice, the incidence of CMV-associated encephalopathy is high, and CMV-targeting treatment bears both anti-CMV and anti-tumor effects. As the major anti-CMV treatment, valganciclovir has demonstrated a promising survival benefit in both newly diagnosed and recurrent glioblastoma as an adjuvant therapy, regardless of surgery and the MGMT promoter methylation state. Immunotherapy, including DC vaccines and adoptive CMV-specific T cells, is also under investigation, and preliminary results have been promising. There are still questions regarding the significance of CMV infection and the carcinogenic mechanism of CMV. Meanwhile, studies have demonstrated the clinical benefits of anti-CMV therapy in glioblastoma. Therefore, anti-CMV therapies are worthy of further recognition and investigation.

## 1. Introduction

Glioblastoma (GBM) is the most common and aggressive tumor in the adult central nervous system. GBM has a dreadful prognosis, despite the combination of surgery, radiotherapy, and chemotherapy used to treat it. The median overall survival (OS) of GBM is approximately 14.4 months, and the 5-year survival rate is below 10% [1,2,3]. GBM represents one of the greatest challenges in the modern era, with a high recurrence rate and a low chance of survival. Therapeutic options are extremely limited at the first diagnosis and relapse. Recent studies have discovered the latent connection between GBM and cytomegalovirus (CMV), shedding light on the possibility of treating GBM with CMV-targeting therapy.

CMV is a double-stranded DNA virus and one of the largest and most complicated kinds of human herpes virus (HHV) [4]. CMV IgG antibodies are found in approximately 60% of adults in developed countries and 100% in developing countries [5]. Notably, the high CMV detection rate does not equate to high CMV activity. Most CMV infections are asymptomatic, establishing a lifelong latent infection. Only in immunosuppressed patients, such as patients with AIDS, organ transplantation patients, and infants [6], will CMV form inclusions in the cell nuclei, perinuclear spaces, and plasma, resulting in cell swelling (cytomegaly) and subsequent CMV disease [5]. CMV infection typically undergoes a three-phase process [7]: the immediate early phase (IE, 0 to 4 h after cell infection), delayed early phase (DE, 4 to 24 h after cell infection), and late phase (>24 h after cell infection). In the IE phase, the proteins IE-72 and IE-86 are expressed to alter the host cell environment and initiate DNA duplication. In the DE phase, non-structural proteins facilitate the viral DNA duplication and adjust the immune responses. Structural proteins, such as the envelope proteins pp65 and gB, are expressed in the late phase and aid in the virus assembly. Proteins, including IE, pp65, and gB, can serve as detection targets for CMV infection.

## 2. Detection of CMV

Currently, there are various methods for detecting CMV infection, including viral DNA detection, antigen and/or serum antibody testing, histopathological evidence, and viral culture. The most widely used indicator of CMV is pp65, which semi-quantitatively reflects the infection activity [8]. Serological tests for IgM and IgG antibodies are less quantitative and require a cut-off value with a higher specificity [9]. Pathology, immunohistochemistry (IHC), in situ hybridization (ISH), polymerase chain reaction (PCR), and tissue microarray (TMA) can also detect the CMV markers [10,11].

Sensitivity varies among different detection methods. A meta-analysis summarized the sensitivity levels of different methods and targets [12]. In CMV-positive patients, the IE protein had the highest association with CMV infection (odds ratio (OR) = 140), followed by the pp65 nucleic acid (OR = 18), pp65 protein (OR = 3.1), and gB nucleic acid (OR = 3.1). With respect to the detection methods, pathology analyses were considered to have high sensitivity. ISH had the highest correlation (OR = 28), polymerase chain reaction (PCR) had an OR of 3.7, and IHC had an OR of 3.5 [12]. The discrepancy between blood serum and whole blood samples should also be noted. The CMV DNA level detected was significantly higher in whole blood samples than that in serum [13]. The establishment of a low-grade infection in glioma cells also provides an explanation as to why CMV was not consistently detected by different groups [14].

## 3. Is CMV Infection Associated with GBM?

CMV has a high detection rate in many primary or metastatic tumors. For instance, in prostate, breast, and colorectal cancer, CMV infection rates are up to 90–100% [15,16,17]. In studies that examined glioma, medulloblastoma, meningioma, and neuroblastoma, CMV pp65 was positive across all tumor types regardless of age, sex, and WHO classification [18,19].

In glioma patients, the CMV detection rate is controversial. Studies have shown that the CMV detection rate is higher in glioma patients than in non-glioma patients. Previous research has suggested that the positive rate of CMV differed more extensively between GBM patients and control patients than those of Epstein Barr virus, human herpesvirus, and herpes simplex virus [20]. CMV DNA is detectable in most tumor samples and blood samples, with a higher detection rate than serum antibodies [11,21,22]. The detection of the CMV genome also confirmed the higher CMV detection rate in GBM patients than in brain tumor or epilepsy patients [23]. With respect to the histopathological evidence, the detection rates of CMV in non-tumor encephalopathies and normal brain samples were significantly lower than those in glioma samples [24]. However, other studies demonstrated that the CMV detection rate among the experimental group was not significantly different from that of the control group [25,26,27]. A meta-analysis summarized the findings of 32 independent studies with over 2000 patients and reported that the CMV detection rate varied extensively due to differences between populations and detection methods [12]. The overall CMV-positive rate in glioma patients was 63% and was significantly associated with gliomas (adjusted OR = 3). The detection rate was not significantly different between different glioma grades [12]. There is no final conclusion about the frequency of CMV in glioma.

## 4. CMV Infection Is Associated with the Prognosis of GBM

It has been reported that a higher expression of the IE protein was correlated with more aggressive tumor progression and shorter OS in extracranial tumors [17]. In GBM patients, positive serological and pathological CMV detection was associated with poor prognosis [28,29], and mainstream scholars agree that patients with higher IE protein expression had significantly shorter OS [24]. A meta-analysis that included seven studies identified no association between the CMV infection and prognosis [30], but among glioma patients with confirmed CMV infection, a low pathological positive rate was associated with better prognosis and longer survival [31,32]. With an IHC cut-off value of 25%, patients with a lower CMV-positive rate had a 20-month longer OS and an 8-month longer PFS than patients with a higher rate. The less positive subgroup also had a significantly higher 2-year survival rate (63.6% vs. 17.2%). However, the difference in median time to progression was non-significant between the lower and higher positive rate patients (for CMV-IEA: 14 vs. 6 months, and for CMV-LA: 8.25 vs. 5 months) [31]. Another case–control study investigated the recursive partition analysis (RPA) subclass, age, surgery, and adjuvant treatments among patients who survived over 18 months, of whom 40% had a low CMV-positive rate, and these patients survived for a median of 42.5 months, indicating that a low-grade CMV infection was strongly associated with long-term survival in GBM patients [32]. Comparatively, among patients with an OS shorter than 18 months, only 8% had a low positive rate, and 47.5% patients had more than 75% of their tumor cells infected [32]. Molecular analyses in CMV-positive rate subgroups have shown that the expression of CMV-IE was significantly associated with p53 mutations, telomerase activity, and several proto-oncogenes, resulting in a more aggressive tumor phenotype [31,33,34,35]. This observation further supports the hypothesis that CMV plays a pathogenetic role in GBM tumors, rather than representing an epiphenomenon, presenting a reasonable explanation for its poor prognosis.

## 5. Mechanisms of CMV-Related Glioma Tumorigenesis

Many preliminary studies have focused on the correlations between CMV and glioma and its tumorigenesis mechanisms from various perspectives. To date, theories regarding tumorigenesis include oncomodulation, oncogenic features, the regulation of the tumor microenvironment, regulation of epithelial-mesenchymal transition (EMT), and the overall regulation of the immune system (Figure 1).

### 5.1. Oncomodulation

Although earlier studies reported that CMV had oncogenic abilities in in vitro experiments and experiments on immunodeficient mice [36], CMV was not able to transform normal human cells. Thus, CMV was considered to have an indirect influence on oncogenesis. Multiple studies have indicated that CMV can interfere with the cell cycle, induce telomerase activation and the DNA damage response, and thus inhibit apoptosis. For instance, the replication stress instigated by CMV infection stimulates the human DNA damage response. Meanwhile, viruses hijack the human stress response transcriptional factors to enhance their own expression of IE72 and IE86. In clinical testing, CMV protein markers are elevated after DNA damage by radiochemotherapy or the recurrence of GBM [37]. CMV is also capable of inducing angiogenesis and cell migration. CMV inhibits cell differentiation and facilitates cancer stem cell preservation. CMV can also induce the expression of oncogenes and inhibit tumor suppressors, such as the p53 mutation. The epigenetic regulation of cell proliferation and immune evasion was also observed in CMV infection [38].

### 5.2. Oncogenic Features

Researchers have discovered that there is a morphological transforming region II imbedded in the CMV genome, which can transform mouse fibroblast cells into malignant cells [39]. The IE protein expressed in the early phase of CMV infection can induce cell transformation through a ‘hit and run’ mechanism, which can activate telomerase to facilitate oncogenesis [40]. Additionally, the IE protein facilitates the correct nuclear localization of the CMV genome during mitosis through the chromatin-tethering domain. The IE protein can also maintain the mitotic cell cycle of the host cells and induce cell proliferation [41]. In addition, CMV expresses the US28 protein, which induces IL-6 expression and STAT3 phosphorylation, both promoting paracrine signaling in oncogenesis [42]. A higher level of VEGF expression is also induced by CMV, which recruits the perivascular cells and promotes angiogenesis [43]. Other studies have indicated that CMV is associated with GSK-3β inhibition and the activation of the WNT, NF-κB, EGFR, ERK, amphiregulin, and SOX-2 pathways [44,45].

### 5.3. Tumor Microenvironment

CMV infection induces COX-2 and 5-LO expression in the tumor microenvironment, leading to the expression of PGE2 and leukotriene as part of inflammatory processes. PGE2 induces cell proliferation and angiogenesis, inhibits apoptosis, and activates invasion, encouraging the formation of the tumor microenvironment [46,47]. CMV infection of the monocytes and neural stem cells generates IL-10 and induces the recruitment of the tumor microenvironment-associated monocytes (macrophages and microglial cells). These monocytes present with the M2 immunosuppressive phenotype, with down-regulated MHC and costimulatory molecules. CMV infection also upregulates B7-H1, an immunosuppressive molecule, which enhances tumor stem cell migration [48].

### 5.4. Epithelial–Mesenchymal Transition

Epithelial–mesenchymal transition (EMT) is an important mechanism of epithelial tumor metastasis. In addition, TGFβ is the lynchpin molecule of the EMT mechanism. CMV induces the expression of TGFβ through multiple pathways [49]. For instance, the IE-72 and IE-86 proteins activate extracellular latent TGFβ1 through matrix metalloproteinase-2 (MMP-2) [49,50]. US28 also facilitates EMT by inducing GSK3β activity. The phosphorylation of GSK3β activates the transcription factors of oncogenes, such as Smads and Snail, to trigger EMT [39]. Additionally, by regulating the IL-10, COX-2, RAS/ERK, and PI3K/AKT pathways, CMV can promote the invasion of tumor cells [51].

### 5.5. Overall Immune System

CMV can also influence the mode of overall immunity. This kind of immune modulation might be related to patient age [52]. In young patients, CMV can generate a broad T-cell receptor pool, enhance the range and ability of T-cell recognition, and enhance anti-tumor immune reactions. The upregulation of CX3CR1 indicates that CMV-specific T cells can recolonize and establish T-cell surveillance. However, in elderly patients, CMV primarily results in T-cell senescence and, therefore, a low survival rate. Unlike PD-1, which can be reversed by checkpoint inhibitors, this senescence is irreversible. Senescent T cells can still access the memory T-cell pool and compete with the binding of specific T-cell receptors, altering the immune reaction and suppressing the overall anti-tumor immune response [53,54].

## 6. CMV-Associated Encephalopathy in Standard Therapy

According to the NCCN guidelines, the standard therapy for GBM includes surgical resection, radiotherapy, and temozolomide (TMZ) chemotherapy [55]. However, studies have reported that radiotherapy can induce the reactivation of CMV, causing CMV-associated encephalopathy, which was defined as a fast neurological decline during the first half course of radiation. According to a case report [56], four patients with sudden, unexpected neurologic decline unrelated to tumor progression during radiotherapy all had high serum CMV DNA copies. Three recovered after ganciclovir or valganciclovir treatment. This suggests that radiotherapy can stimulate CMV reactivation. A prospective observational study [57], GLIO-CMV-01, included 50 patients (27 with metastatic brain tumors and 23 with high-grade gliomas). Among the patients with positive serological CMV antigens prior to treatment, 48% experienced CMV viremia, and 87% experienced CMV-associated encephalopathy that required active treatment. No CMV-associated encephalopathy occurred among the CMV serology-negative patients. In addition to surgery and radiochemotherapy, many GBM patients also receive glucocorticoids in the perioperative period. Studies have shown that glucocorticoids may be linked to the reactivation of CMV [57,58].

CMV-associated encephalopathy is associated with poor prognosis. It has been reported that, at 150 days after radiotherapy, 74% (14/19) of patients without CMV encephalopathy were still alive, while only 54% (7/13) of patients with CMV encephalopathy were still alive. Meanwhile, the use of ganciclovir or valganciclovir mitigated the encephalopathy symptoms and improved the prognosis [57]. Another study demonstrated that, in 118 brain tumor patients with sudden neurologic decline during radiotherapy, 24% had CMV viremia. Among GBM patients, the OS was 99 days in the encephalopathy group compared with 570 days in the unaffected group. For NSCLC brain metastases, the OS was 47 versus 219 days in the encephalopathy group and the unaffected group, respectively. Further analysis showed that a low basophilic granulocyte count before treatment was associated with a higher risk of CMV-associated encephalopathy [59]. The high incidence of CMV-associated encephalopathy and poor survival time suggest that researchers should carry out clinical treatment targeting CMV.

## 7. Anti-CMV Therapy in GBM

Many studies have suggested that anti-CMV therapy can restrain glioma progression in vitro and in vivo. Relevant clinical trials have been promoted (Table 1 and Table 2). There are four major treatment strategies for targeting CMV: valganciclovir, dendritic cell vaccine, adoptive CMV-specific T-cell therapy, and peptide vaccine (Figure 2).

### 7.1. Valganciclovir

Valganciclovir inhibits viral DNA duplication by competing with deoxyguanosine triphosphate to bind the DNA polymerase. Randomized controlled trials indicated that valganciclovir was effective in treating CMV disease and solid organ transplant recipients [71]. Anti-CMV drugs were able to suppress medulloblastoma and neuroblastoma xenografts in vivo by 72% and 40%, respectively [57,72]. Valganciclovir has been proven to be effective in CMV-positive medulloblastoma, while CMV-negative tumors do not benefit from this treatment [73].

The efficacy of valganciclovir in GBM is of great interest. The most relevant studies are the Valcyte Treatment of Glioblastoma Patients in Sweden (VIGAS) trial and its follow-up studies. VIGAS is a double-blind, randomized controlled trial that included 42 newly diagnosed GBM patients and compared the impacts of standard treatment with or without a standard dose of valganciclovir until the disease progression or voluntary withdrawal [60]. In the safety assessment, valganciclovir was safe and well tolerated in parallel with chemoradiotherapy. The primary endpoint (reduced tumor volume on MRI at 6 months, PFS, OS) was unfortunately not reached. Among patients who had taken valganciclovir for at least 6 months, the survival benefit was more significant. The median OS of patients with at least 6 months of valganciclovir treatment was 24.1 months, versus 13.1 months in patients with a shorter treatment duration and 13.7 months in the control group [60,62]. However, a subsequent article from the same research group reported significant controversy and publicity. A further exploratory analysis enrolled 28 patients receiving valganciclovir, with a median OS of 25.0 months compared with 13.5 months in the contemporary controls and a 2-year OS rate of 62% versus 18% [63]. After further improvement of the analytical methods, valganciclovir was still associated with survival benefits, and the sustained valganciclovir subgroup had prolonged survival [61]. This study has been heavily criticized based on its unjustified patient selection bias, the limited characterization of the control group, the mathematical assessment of the data analysis, and the low detection rate of CMV infection [74,75,76,77]. These critical issues severely lessened the validity of the VIGAS trial, which provided limited evidence for the effects of valganciclovir therapy. The same research team then reviewed 102 newly diagnosed GBM patients and discovered that OS was positively correlated with valganciclovir treatment, complete resection, and the methylated MGMT promoter [64].

Valganciclovir treatment in secondary and recurrent GBM also showed promising results. Among 44 secondary GBM patients, 8 of them used valganciclovir as an add-on to secondary therapy. The OS after recurrence improved from 12.7 months to 19.1 months and the 24-month OS rates increased from 2.8% to 37.5% [78]. The latest retrospective study included 29 recurrent GBM patients, who used valganciclovir as an add-on. The dose was well tolerated and prolonged OS from 7.4 months to 12.1 months, compared with the control group, irrespective of MGMT methylation status or re-operation. However, neither of these two retrospective studies tested for CMV infection or viral load. They also included widely varying second-line therapies, including reoperation, hypofractionated radiation therapy, temozolomide, lomustine (CCNU), bevacizumab, and/or gamma knife treatment [79].

Another team reported that the combination of valganciclovir and bevacizumab exhibited a better survival trend in recurrent GBM patients. The subgroup of concurrent valganciclovir and bevacizumab had a 6-month PFS (PFS-6) of 50% and a median OS of 11.3 months, which were significantly longer than those of the historical controls [80]. An ongoing phase-II randomized controlled multi-center trial using valganciclovir is awaited.

### 7.2. Dendritic Cell Vaccine

CMV can be specifically recognized by the T cells or MHC as foreign matter. It is enriched in tumor tissue yet sparsely scattered in the peripheral blood and other tissue. Preclinical studies have predicted the potential treatment effects of immunotherapy targeting CMV-specific immune cells [81,82]. CMV-specific dendritic cell (DC) vaccines usually target the pp65 antigen. The first phase-Ⅰ clinical trial, ATTAC-GM, used a DC vaccine with dose-intensive TMZ and GM-CSF and showed preliminary clinical benefits [65]. The median OS was 37.7 months, and the survival rate at 60 months was 36.4%. A total of 4 out of 11 patients remained stable for 59–64 months after diagnosis. Another ATTAC trial compared the effects of tetanus toxoid preconditioning on a DC vaccine with standard TMZ [66]. The median OS was 38.3 months in the experimental group, with an 18-month OS of 66.7% and a 60-month OS of 33.3%. The DC-only group reached a median OS of 13.9 months. ATTAC trials showed that the immune adjuvant tetanus toxoid combined with the DC vaccine is a promising therapy. Using tetanus toxoid preconditioning before DC vaccination promotes DC cell migration, stimulates the immune response, and is associated with prolonged survival [83]. It has been reported that one-third of GBM patients who have received CMV-specific DC vaccines have shown surprisingly elongated survival [84]. Multiple clinical trials are being initiated to study the role of tetanus diphtheria (NCT03615404, NCT03927222, NCT02465268, NCT02366728, NCT03688178, NCT00693095).

### 7.3. Adoptive CMV-Specific T Cells

Represented by CAR-T, adoptive cell therapy (ACT) targeting CMV has also been studied recently. CMV-specific CD8 T cells target tumor cells directly and stimulate the cytokines, NK cells, macrophages, and memory T cells indirectly. A prospective phase-Ⅰ study with 25 newly diagnosed GBM patients treated with CAR-T therapy yielded a median OS of 21 months and a PFS of 10 months [68]. The subgroup analysis suggested that patients receiving CAR-T therapy before recurrence had a median OS of 23 months, in comparison with patients who received CAR-T after recurrence, whose median OS was 14 months. The T-cell gene signature was associated with improved long-term survival [68]. Another phase-Ⅰ/Ⅱ clinical trial included 65 newly diagnosed or recurrent GBM patients whose OS, PFS, and PFS-6 were 12 months, 1.3 months, and 19%, respectively. Further study demonstrated that the DC vaccine ATTAC can enhance the polyfunctionality of ACT therapy and increase overall survival, indicating its potential for combination immunotherapy [70]. Generally, the effect of T cells is attenuated and does not revert the immunologically inhibitory microenvironment. Researchers also noted that patients with CMV-positive blood serum samples were not always sensitive to CMV-specific T cells. The underlying mechanism requires future study [69]. Further clinical trials are ongoing (NCT01109095, NCT00693095).

## 8. Conclusions

Studies have suggested that the CMV infection rate is high in glioma patients, which is associated with survival prognosis. As an immunosuppressive microenvironment in glioma, CMV is reactivated and then promotes tumorigenesis through various strategies [56,85]. Anti-CMV treatments, such as valganciclovir, have demonstrated promising survival benefits in glioma. Patients may benefit from adjuvant anti-CMV treatment as part of comprehensive therapy. Immunotherapy is mostly carried out in the early stage of clinical trials; however, the immunotherapy of DC vaccines and adoptive CMV-specific T cells has displayed promising results. In general, anti-CMV treatment shows potential and is a treatment mode deserving of clinicians’ attention. Future clinical research should not only focus on the CMV infection rate or the relationship between CMV infection and tumorigenesis, but also anti-CMV treatments and combination therapies (e.g., glucocorticoids and radiotherapy).

## Figures and Tables

**Figure 1 jcm-11-05221-f001:**
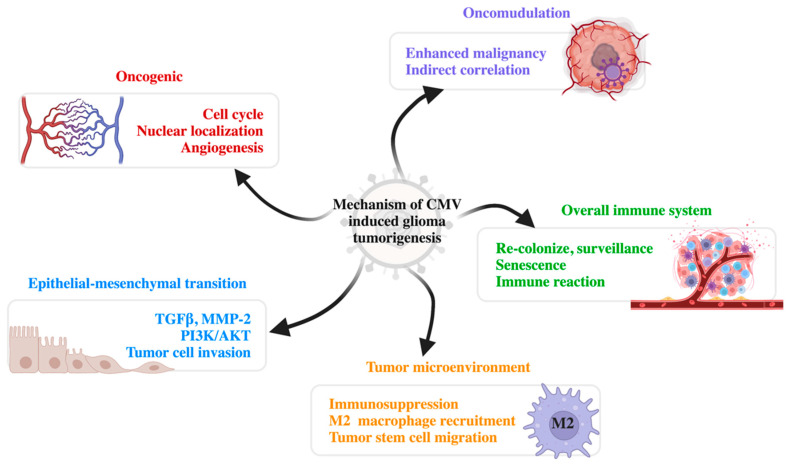
Mechanisms of CMV-induced glioma tumorigenesis. There are five major mechanisms of CMV tumorigenesis: oncogenic, oncomodulation, epithelial-mesenchymal transition, modified tumor microenvironment, and influence on the overall immune system. CMV, cytomegalovirus; TGFβ, transforming growth factor β; MMP-2, matrix metalloproteinase-2; PI3K/AKT, phosphatidylinositol 3-kinase/protein kinase B (PKB).

**Figure 2 jcm-11-05221-f002:**
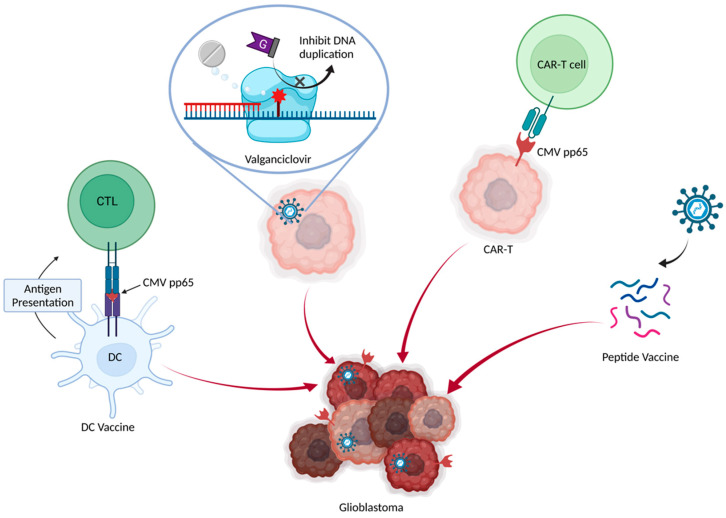
Various pathways of CMV-related treatment of glioblastoma. There are four major treatment methods targeting CMV. Valganciclovir is an anti-CMV drug, which also shows potential survival benefits for GBM. Immune therapy can exploit the CMV pp65 antigen as a specific target for the tumor cells. CAR-T cells can be engineered to target the CMV pp65 antigen, directly killing the tumor. Dendritic cells primed by the CMV pp65 antigen serve as a dendritic cell vaccine to recruit the cytotoxic T lymphocytes. CMV can also be used to create peptide vaccines. CMV, cytomegalovirus; DC, dendritic cell; CAR-T chimeric antigen receptor T cell; CTL, cytotoxic T lymphocyte.

**Table 1 jcm-11-05221-t001:** CMV-related clinical studies to treat GBM, with published articles.

Treatment	Year	Trial Registration	Trial Name	Interventions	Patients	N	Phase	Outcome	Reference
Valganciclovir	2013	NCT00400322	VIGAS	Valganciclovir + CRT vs. CRT	ND-GBM	42	I/II	No difference in tumor volume, OS, and PFS. OS-24 27.3% in experiment arm vs. 25% in control. Treatment >6 m OS 24.1 m, <6 m OS 13.1 m. Treatment >6 m OS-2 year 67.6%, OS-4 year 27.3%	Soderberg-Naucler, C. [60]
	2013	-	VIGAS re-analysis	Valganciclovir + CRT vs. CRT	ND-GBM		retrospective	Valganciclovir vs. control HR 2.44, treatment >6 m vs. control HR 0.441, treatment >6 m vs. <6 m HR 1.351	Soderberg-Naucler, C. [61]
	2014	-	VIGAS re-analysis	Valganciclovir + CRT vs. CRT	ND-GBM		retrospective	Patients with lower viral load have better prognosis	Malte Ottenhausen [62]
	2013	-	VIGAS further study	Valganciclovir + CRT vs. CRT	ND-GBM	50	retrospective	OS-24 62% in experiment arm vs. 18% in control; OS 25.0 m vs. 13.5 m. Treatment >6 m OS-24 70%, OS 30.1 m	Soderberg-Naucler, C. [63]
	2020	-		Valganciclovir + CRT vs. CRT	R-GBM	8	retrospective	OS after relapse 19.1 m in experiment arm vs. 12.7 m in control; OS-24 37.5% vs. 2.8%	Soderberg-Naucler, C. [64]
DC vaccine	2017	NCT00639639	ATTAC-GM	DI-TMZ + GM-CSF + CMV pp65 RNA-pulsed DC vs. CRT	ND-GBM	11	I	PFS 25.3 m, OS 41.1 m, 4 patients survived longer without progression (59–64 m)	John H. Sampson [65]
	2015	NCT00639639	ATTAC	CMV pp65 RNA-pulsed DC + Td+ CRT vs. unpulsed DC + Td+ CRT vs. CRT	ND-GBM	12	I	PFS, OS no worse than control, 3 patients survived >36.6 m. Td enhances DC vaccine because CCL3 enhances DC migration and inhibits tumor progression	John H. Sampson [66]
CAR-T	2014	ACTRN12609000338268		Autologous CMV pp65-specific T cells	R-GBM	19	I	PFS 246 d, OS 403 d, 4 of 10 patients remained progression-free during study	Rajiv Khanna [67]
	2020	ACTRN12615000656538		Autologous CMV pp65-specific T cells	ND-GBM	25	I	PFS 25 m, PFS-12 20%, OS 21 m, OS-12 52%. Treatment before relapse was significantly longer OS than that after relapse	Rajiv Khanna [68]
	2020	NCT02661282		Autologous CMV pp65-specific T cells	ND + R-GBM	65	I/II	Increased circulating CMV-CD8 T cells, but did not improve survival	Amy B Heimberger [69]
	2017	NCT00693095.	ATCT	CMV pp65-specific T cells + CMV pp65 RNA-pulsed DC vs. CMV pp65-specific T cells	ND-GBM	22	I	CMV DC vaccine enhanced polyfunctionality of adoptive CMV-specific T cells, correlated with OS.	John H. Sampson [70]

N, number; CRT, standard chemoradiotherapy; ND, newly diagnosed; R, recurrent; GBM, glioblastoma; PFS, progression-free survival; OS, overall survival; OS-24/-5, 24 month/5 month overall survival; HR, hazard ratio; m, month(s); d, day(s); TMZ, temozolomide; DI-TMZ dose-intensified temozolomide; DC, dendritic cell; Td, tetanus-diphtheria; GBM, glioblastoma; HGG, high-grade glioma.

**Table 2 jcm-11-05221-t002:** Ongoing clinical trials of anti-CMV treatment in GBM.

Treatment	Research Team	Trial Registration	Year	Trial Name	Study Title	Treatment Plan	Patients	N	Phase	Status
Valganciclovir	Cecilia Soderberg-Naucler	NCT04116411	September 2019–August 2024	VIGAS2	A Clinical Trial Evaluating the Efficacy of Valganciclovir in Glioblastoma Patients	Valganciclovir + CRT vs. placebo + CRT	ND-GBM	220	II, Randomized	recruiting
DC vaccine	Gary Archer	NCT03615404	October 2018–July 2020	ATTAC-P	Cytomegalovirus (CMV) RNA-Pulsed Dendritic Cells for Pediatric Patients and Young Adults With WHO Grade IV Glioma, Recurrent Malignant Glioma, or Recurrent Medulloblastoma	DI-TMZ + GM-CSF + Td + CMV pp65 RNA-pulsed DC	ND + R-GBM, recurrent medulloblastoma	11	I	completed
	Gary Archer	NCT03927222	September 2019–December 2023	I-ATTAC	Immunotherapy Targeted Against Cytomegalovirus in Patients With Newly-Diagnosed WHO Grade IV Unmethylated Glioma	DI-TMZ + GM-CSF + Td+ CMV pp65 RNA-pulsed DC	ND-GBM	48	II	recruiting
	Duane Mitchell	NCT02465268	August 2016–June 2024	ATTAC-II	Vaccine Therapy for the Treatment of Newly Diagnosed Glioblastoma Multiforme	GM-CSF + Td + CMV pp65 RNA-pulsed DC vs. un-pulsed PBMC	ND-GBM	120	II, Randomized	recruiting
	Gary Archer	NCT02366728	October 2015–August 2020	ELEVATE	DC Migration Study for Newly-Diagnosed GBM	DC + CMV pp65 RNA-pulsed DC + TMZ vs. Td + CMV pp65 RNA-pulsed DC + TMZ vs. basiliximab + Td+ CMV pp65 RNA-pulsed DC + TMZ	ND-GBM	100	II, Randomized	Active, not recruiting
	Gary Archer	NCT03688178	August 2020–March 2025	DERIVe	DC Migration Study to Evaluate TReg Depletion In GBM Patients With and Without Varlilumab	DC pre-conditioning vaccine + TMZ vs. Td pre-conditioning + DC vaccine + TMZ vs. DC Vaccine + varlilumab (Td pre-conditioning) + TMZ	ND + R-GBM	112	II, Randomized	recruiting
DC vaccine+ CAR-T	John Sampson	NCT00693095	September 2008–April 2015	ERaDICATe	Evaluation of Recovery From Drug-Induced Lymphopenia Using Cytomegalovirus-specific T-cell Adoptive Transfer	CMV-autologous lymphocyte transfer (CMV-ALT) vs. CMV-ALT + CMV pp65 RNA-pulsed DC	ND-GBM	23	I, Randomized	completed
CAR-T	Nabil Ahmed	NCT01109095	October 2010–March 2018	HERT-GBM	CMV-specific Cytotoxic T Lymphocytes Expressing CAR Targeting HER2 in Patients With GBM	HER2-CAR CMV-specific CTL	R-GBM	16	I	completed
Peptide Vaccine	Gary Archer	NCT02864368	December 2016–September 2021	PERFORMANCE	Peptide Targets for Glioblastoma Against Novel Cytomegalovirus Antigens	PEP CMV + Td + CRT vs. PEP CMV + Td+ TMZ	ND-GBM	70	I, Randomized	Active, not recruiting
Observational	Benjamin Frey	NCT02600065	November 2015–February 2020	GLIO-CMV-01	Analysis of CMV Infections in Patients With Brain Tumors or Brain Metastases During and After Radio(Chemo)Therapy	CRT + TMZ	HGG, metastases	250	Observation	recruiting

N, number; CRT, standard chemoradiotherapy; ND, newly diagnosed; R, recurrent; GBM, glioblastoma; TMZ, temozolomide; DI-TMZ dose-intensified temozolomide; Td, tetanus-diphtheria; DC, dendritic cell; y/o, years old; PBMC, peripheral blood mononuclear cell; PEP, peptide; HGG, high-grade glioma. Trials were searched for on the website: clinicaltrials.gov until 1 July 2022.

## Data Availability

Not applicable.

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
