# Peer review of "Cytomegalovirus and Glioblastoma: A Review of the Biological Associations and Therapeutic Strategies"

_jcm, 2022, doi:10.3390/jcm11175221_

Round 1

Reviewer 1 Report

I congratulate with Authors for this well written review. I believe this is of interest for many readers.

I have only few questions or critical points for Authors:

1) “Although a meta-analysis that included 7 studies showed no association between CMV infection and prognosis24, mainstream scholars agree that patients with higher IE protein expression had significantly shorter OS18”  This sentence should be reformulated as in the pyramid of evidence meta-analysis are far above expert opinions.

Likewise, in the Conclusion session, the sentence “Scholars have now established that CMV is associated with glioma occurrence and progression” is not in line with the logical flow of the period. It seems that some opinions are stronger than other evidences, which are more controversial (CMV infection is the cause of the consequence?)

2) What is the rate of silent CMV infection in healthy population? Can we accurately assess if a silent/dormant CMV occurred if there is no active infection/reactivation.

Indeed, a number of glioma patients are never found positive to CMV. Is this due to a lack of sensitivity of our detection methods?

Moreover, we understand from present study that survival is different between CMV-positive and CMV-negative patients, with some differences also among CMV-positive patients according to their lower or higher positivity scores. Are there any biological differences (molecular, genomic, epigenomic, etc.) between this subgroups of patients? 

Is the impressive difference in OS due to different tumor characteristics or CMV reactivation plays a role in patients poor survival independently from the brain tumor? Namely, do they die because of a different tumor relapse characteristics (i.e. more rapid, more diffused, bigger volumes, etc.) or because of less defined neurological or non-neurological causes like the CMV-encephalopathy that has been mentioned?

Lastly, glioblastoma patients are obviously immunodeficient patients due to both the pathology and the treatment, what is the CMV infection/reactivation course in other immunosuppressed patients (AIDS, transplanted, other malignant tumors, etc.)? Is there a difference with their CMV-negative counterpart? Again, if they all share a similar more aggressive course, it could independently explain the worse outcome of GBM CMV-positive cases.

Author Response

Dear Editors and Reviewers:

Thank you for your advice and comments on our manuscript jcm-1824855 (Title: Cytomegalovirus and Glioblastoma: A Review of Biological Association and Therapeutic Strategies). According to your comments, we have made significant modifications to our manuscript to ensure our review is more accurate and convincing. We have provided a revised manuscript edited using the ‘Track Changes’ tool in Microsoft Word, with the changes in response to the reviewers’ comments highlighted, while the language changes were not highlighted. A clean copy of the revised manuscript that incorporates all changes is also provided. The detailed point-by-point responses are listed as follows:

<Responses to reviewer #1’s comments>

Comment 1: “Although a meta-analysis that included 7 studies showed no association between CMV infection and prognosis, mainstream scholars agree that patients with higher IE protein expression had significantly shorter OS18” This sentence should be reformulated as in the pyramid of evidence meta-analysis are far above expert opinions.

Likewise, in the Conclusion session, the sentence “Scholars have now established that CMV is associated with glioma occurrence and progression” is not in line with the logical flow of the period. It seems that some opinions are stronger than other evidences, which are more controversial (CMV infection is the cause of the consequence?)

Changes: Reformulate sentence ‘In GBM patients, positive serological and pathological CMV detection was associated with poor prognosis, and mainstream scholars agree that patients with higher IE protein expression had significantly shorter OS. A meta-analysis that included 7 studies showed no association between CMV infection and prognosis, but among glioma patients with confirmed CMV infection, a low pathological positive rate was associated with better prognosis and longer survival’ in lines 123-125. Modify the ‘conclusion’ section to ‘Studies have suggested that the CMV infection rate is high in glioma patients, which is associated with survival prognosis. As immunosuppressive microenvironment in glioma, CMV is reactivated and then promotes tumorigenesis by various strategies53,81. Anti-CMV treatment, such as valganciclovir, has shown promising survival benefit in glioma. Patients may benefit from adjuvant anti-CMV treatment to comprehensive therapy. Immunotherapy is mostly in the early stage of clinical trials, however, immunotherapy of DC vaccines and adoptive CMV-specific T cells has displayed promising results. In general, anti CMV treatment has potential and is a treatment mode commendable of clinicians' attention. Future clinical research should not only focus on the CMV infection rate or the relationship between CMV infection and tumorigenesis, but also anti-CMV treatment and combination therapies (e.g., glucocorticoids and radiotherapy)’ in lines 346-367.

Responses: Thank you for your suggestions. We rearranged the sentence order based on the strength of evidence. While most studies indicated that the effect between CMV and glioma is bidirectional, further studies are still in need. We modified the conclusion section to convey more clearly. In general, anti CMV treatment has great potential and is a treatment mode commendable of clinicians' attention. At present, many clinical trials are being carried on, and the results are worthy of expectation and promotion. The purpose of this review is also to improve researchers' understanding of CMV in glioma, promote the research of anti CMV therapy, and improve patients' benefits.

Comment 2: What is the rate of silent CMV infection in healthy population? Can we accurately assess if a silent/dormant CMV occurred if there is no active infection/ reactivation.

Indeed, a number of glioma patients are never found positive to CMV. Is this due to a lack of sensitivity of our detection methods?

Moreover, we understand from present study that survival is different between CMV-positive and CMV-negative patients, with some differences also among CMV-positive patients according to their lower or higher positivity scores. Are there any biological differences (molecular, genomic, epigenomic, etc.) between these subgroups of patients?

Is the impressive difference in OS due to different tumor characteristics or CMV reactivation plays a role in patients’ poor survival independently from the brain tumor? Namely, do they die because of a different tumor relapse characteristics (i.e. more rapid, more diffused, bigger volumes, etc.) or because of less defined neurological or non-neurological causes like the CMV-encephalopathy that has been mentioned?

Lastly, glioblastoma patients are obviously immunodeficient patients due to both the pathology and the treatment, what is the CMV infection/reactivation course in other immunosuppressed patients (AIDS, transplanted, other malignant tumors, etc.)? Is there a difference with their CMV-negative counterpart? Again, if they all share a similar more aggressive course, it could independently explain the worse outcome of GBM CMV-positive cases.

Changes: Add sentence ‘CMV IgG antibodies are found in approximately 60% of adults in developed countries and 100% in developing countries. Notably, high CMV detection rate does not represent high CMV activity. Most CMV infections are asymptomatic, establishing lifelong latent infection’ in lines 63-65. Add sentence ‘For detection methods, pathology analyses were considered with high sensitivity’ in lines 87-88. Add sentence ‘CMV DNA level was detected significantly higher in whole blood samples than serum. The establishment of low-grade infection in glioma cells also provides an explanation why CMV was not consistent detected by different groups’ in lines 90-92. Add sentence ‘There is no final conclusion about the frequency of CMV in glioma’ in lines 111-112.Add sentence ‘However, the difference of median time to progression was non-significant between lower and higher positive rate patients (for CMV-IEA: 14 vs. 6 months, for CMV-LA: 8.25 vs. 5 months). Another case-control study adjusted recursive partition analysis (RPA) subclass, age, surgery and adjuvant treatments’ in lines 130-133. Add phrase ‘indicating that low-grade CMV infection was strongly associated with long-term survival in GBM patients’ in lines 134-135. Add sentence ‘Molecular analyses in CMV positive rate subgroups shown that expression of CMV-IE was significantly associated with p53 mutations, telomerase activity, and several protooncogenes, resulting in a more aggressive tumor phenotype. This observation further supports the hypothesis that CMV may play a pathogenetic role in GBM tumors rather than representing an epiphenomenon, presenting a reasonable explanation for poor prognosis’ in lines 137-141.

Responses: Thank you for your professional comments and we have improved the manuscript as shown above. Most CMV infections are asymptomatic in immune non-deficient population. The sensitivity of different detection targets, detection methods and sample types are indeed different, which may be one of the reasons for the low infection rate in some studies. Detection method should be unified among different research groups. As for patients who have never found CMV positive after using multiple tests, they might be free of CMV infection.

For survival prognosis in low- and high-grade infection patients, Rahbar et al. published that after adjusting recursive partition analysis (RPA) subclass, age, surgery and adjuvant treatments (except for gamma-knife), CMV low-infection grade was significantly associated with longer overall survival and longer time to progression as an independent prognosis factor.  At the same time, age, gender, extent of surgery and gamma-knife treatment were found non-significantly associated with survival. Furthermore, Expression of CMV-IE was significantly associated with p53 mutations, telomerase activity, and several protooncogenes. Thus, the shortening of survival caused by high infection of CMV may be partly explained by its tumorigenicity. Further differences in molecules, genomes and epigenetics have not been reported yet. Reviewer raised a very scientific question, which are worthy of follow-up exploration.

As for the cause of death of CMV positive patients, according to Rahbar's report, there was no significant difference in treatment of patients with high or low infection except for gamma knife. CMV reactivation and encephalopathy during radiotherapy and chemotherapy are also important reasons for poor prognosis. However, the clinical manifestations of tumors, such as growth rate, size and location, have not been reported in detail, which is worthy of follow-up investigation.

CMV is considered one of the most common opportunistic pathogens in immunocompromised patients, including AIDS patients, transplant recipients, and developing fetuses, resulting in CMV disease. The direct clinical results of CMV disease come from virus replication, transmission, and tissue invasion, leading to viremia. However, in tumor patients, CMV mainly regulates gene expression, transforming normal cells into tumor cells (described in the article), rather than the massive expansion of the virus into the blood. Therefore, the mechanism is different. The pathogenic mechanism and treatment strategies in immunosuppressors have been extensively studied, but in tumor, especially gliomas, details still need to be explored, which is also the focus of our review.

Reviewer 2 Report

Thanks for the opportunity of reading the manuscript of the review (jcm-1824855) entitled “Cytomegalovirus and glioblastoma: a review of biological association and therapeutic strategies”. The authors performed a literature review to investigate the epidemiology of CMV, the mechanisms of tumorigenesis, and its therapeutic strategies in GBM.

While this study seems comprehensive, the review as it is fails to give the reader a clear answer on therapeutic potential of anti-CMV treatment. The authors need to clarify the overall message they are trying to convey. 

According to the instructions of the submission, all figures and tables should be inserted near the first citation in the main text. I recommend that Figures 1, Tables 1 and Table 2 should be inserted in the main text, not in the conclusion section.

Tables 1 and 2 are difficult to read and understand. It needs to be improved to make it easier for the reader to understand. In particular, the reference numbers should be included in each table so that the reader can easily understand which is the source of the citation. 

In the abstract the authors write that CMV plays a crucial role in the pathogenesis and treatment of GBM. Another figure of the specific roles of CMV in GBM would be very helpful for understanding for the readers.

Since this is a review article, references should be clearly stated. It is noticeable that the text describing known information is not accompanied by a reference. For example, “Introduction” section, line 8, “Detection of CMV” section, line 6 and 12, “Is CMV infection associated with GBM?” section, line 6 and 19 and “CMV infection is associated with the prognosis of GBM” section, line 13, etc.

In addition, you should state how you searched for and collected the citations.

Author Response

Dear Editors and Reviewers:

Thank you for your advice and comments on our manuscript jcm-1824855 (Title: Cytomegalovirus and Glioblastoma: A Review of Biological Association and Therapeutic Strategies). According to your comments, we have made significant modifications to our manuscript to ensure our review is more accurate and convincing. We have provided a revised manuscript edited using the ‘Track Changes’ tool in Microsoft Word, with the changes in response to the reviewers’ comments highlighted, while the language changes were not highlighted. A clean copy of the revised manuscript that incorporates all changes is also provided. The detailed point-by-point responses are listed as follows:

<Responses to reviewer #2’s comments>

Comment 1: While this study seems comprehensive, the review as it is fails to give the reader a clear answer on therapeutic potential of anti-CMV treatment. The authors need to clarify the overall message they are trying to convey.

Changes: Delete sentence ‘Therefore, it is debatable whether the immunosuppression of GBM leads to positive CMV detection or whether CMV activation induces glioma. Most studies believe that this effect is bidirectional’. Move sentence ‘Notably, high CMV detection rate does not represent high CMV activity’ to lines 64-65. Modify the ‘conclusion’ section to ‘Studies have suggested that the CMV infection rate is high in glioma patients, which is associated with survival prognosis. As immunosuppressive microenvironment in glioma, CMV is reactivated and then promotes tumorigenesis by various strategies53,81. Anti-CMV treatment, such as valganciclovir, has shown promising survival benefit in glioma. Patients may benefit from adjuvant anti-CMV treatment to comprehensive therapy. Immunotherapy is mostly in the early stage of clinical trials, however, immunotherapy of DC vaccines and adoptive CMV-specific T cells has displayed promising results. In general, anti CMV treatment has potential and is a treatment mode commendable of clinicians' attention. Future clinical research should not only focus on the CMV infection rate or the relationship between CMV infection and tumorigenesis, but also anti-CMV treatment and combination therapies (e.g., glucocorticoids and radiotherapy)’ in lines 346-367.

Responses: Thank you for your professional suggestions. We modified the conclusion part to convey more clearly. With the gradual deepening of the understanding of CMV, the treatment of anti CMV has gradually been paid attention. Gratifying effects of valganciclovir have been reported. Compared with other immunotherapy methods, CMV is considered potentially for prolonging survival. In general, anti CMV treatment has great potential and is a treatment mode commendable of clinicians' attention. At present, many clinical trials are being carried on, and the results are worthy of expectation and promotion. The purpose of this review is also to improve researchers' understanding of CMV in glioma, promote the research of anti CMV therapy, and improve patients' benefits.

Comment 2: According to the instructions of the submission, all figures and tables should be inserted near the first citation in the main text. I recommend that Figures 1, Tables 1 and Table 2 should be inserted in the main text, not in the conclusion section.

Changes: Insert figure 1 in ‘Mechanisms of CMV related glioma tumorigenesis’ section in line 147. Add paragraph ‘Many studies have suggested that anti-CMV therapy can restrain glioma progression in vitro and in vivo. Relevant clinical trials are promoted (Table 1 and Table 2). There are four major treatment strategies inspired by CMV: valganciclovir, dendritic cell vaccine, adoptive CMV-specific T-cell, and peptide vaccine (Figure 2)’ in ‘anti-CMV therapy in GBM’ section in lines 242-245.

Responses: Thank you for your careful checks and comments. We inserted Figure 1(new figure showing tumorigenesis mechanisms), Figure 2(major treatment strategies), Table 1 and Table 2 in the main text, after the first citation.

Comment 3: Tables 1 and 2 are difficult to read and understand. It needs to be improved to make it easier for the reader to understand. In particular, the reference numbers should be included in each table so that the reader can easily understand which is the source of the citation. 

Changes: Modify Table 1 and Table 2 and insert them in lines 260-264.

Responses: Thank you for your suggestions. We have improved the tables more clear to read. Reference was labeled in brackets in previous version, we uniformed the reference format of tables with the manuscript.

Comment 4: In the abstract the authors write that CMV plays a crucial role in the pathogenesis and treatment of GBM. Another figure of the specific roles of CMV in GBM would be very helpful for understanding for the readers.

Changes: Add Figure 1 in lines 157-162, after the first citation.

Responses: Thank you for your professional comments. We summarized the mechanisms of CMV induced glioma tumorigenesis in Figure 1 to help understanding.

Comment 5: Since this is a review article, references should be clearly stated. It is noticeable that the text describing known information is not accompanied by a reference. For example, “Introduction” section, line 8, “Detection of CMV” section, line 6 and 12, “Is CMV infection associated with GBM?” section, line 6 and 19 and “CMV infection is associated with the prognosis of GBM” section, line 13, etc.

Changes: Add reference 4 in line 63. Add reference 9 in line 82. Add reference 12 in line 109 and line 111. Add reference 32 in line 135. Add relevant references in corresponding sites when describing published studies.

Responses: Thank you for your careful checks. We have stated the references in corresponding sites, and we checked the reference citations in overall manuscript.

Comment 6: In addition, you should state how you searched for and collected the citations.

Responses: Thank you for your proposal. A total of 85 citations are included in this manuscript.

Published studies were searched on website pubmed.gov until 1st July 2022. 359 results were searched with the search term ‘cytomegalovirus & glioblastoma’, and 287 articles were searched with the search term ‘cytomegalovirus & glioblastoma. In sections of virus infection and pathogenic mechanism, we cited the research based on publication time, impact factors and article quality, mainly original article and meta-analysis, and included some representative reviews. Due to the large number of studies, our review only selects the representative ones for reference.

In treatment section, we cited all clinical studies, including clinical trials, cohort studies, case report, case series, and letters published on the PubMed website, to introduce all currently published studies. By using ‘cytomegalovirus & glioblastoma’ as the search term and filtered ‘clinical trial’ in PubMed website, a total of 10 articles were searched.

  1. ‘Stragliotto G, Rahbar A, Solberg NW, et al. Effects of valganciclovir as an add-on therapy in patients with cytomegalovirus-positive glioblastoma: a randomized, double-blind, hypothesis-generating study. Int J Cancer. 2013;133(5):1204-1213.’ was cited as reference 60.
  2. ‘Batich KA, Reap EA, Archer GE, et al. Long-term Survival in Glioblastoma with Cytomegalovirus pp65-Targeted Vaccination. Clin Cancer Res. 2017;23(8):1898-1909.’ was cited as reference 65.
  3. ‘Schuessler A, Smith C, Beagley L, et al. Autologous T-cell therapy for cytomegalovirus as a consolidative treatment for recurrent glioblastoma. Cancer Res. 2014;74(13):3466-3476.’ was cited as reference 67.
  4. ‘Smith C, Lineburg KE, Martins JP, et al. Autologous CMV-specific T cells are a safe adjuvant immunotherapy for primary glioblastoma multiforme. J Clin Invest. ’ was cited as reference 68.
  5. ‘Weathers SP, Penas-Prado M, Pei BL, et al. Glioblastoma-mediated Immune Dysfunction Limits CMV-specific T Cells and Therapeutic Responses: Results from a Phase I/II Trial. Clin Cancer Res. 2020;26(14):3565-3577.’ was cited as reference 69.
  6. ‘Reap EA, Suryadevara CM, Batich KA, et al. Dendritic Cells Enhance Polyfunctionality of Adoptively Transferred T Cells That Target Cytomegalovirus in Glioblastoma. Cancer Res. 2018;78(1):256-264.’ was cites as reference 70.
  7. ‘Mitchell DA, Batich KA, Gunn MD, et al. Tetanus toxoid and CCL3 improve dendritic cell vaccines in mice and glioblastoma patients. 2015;519(7543):366-369.’ was cited as reference 66.
  8. ‘Batich KA, Mitchell DA, Healy P, Herndon JE, Sampson JH. Once, Twice, Three Times a Finding: Reproducibility of Dendritic Cell Vaccine Trials Targeting Cytomegalovirus in Glioblastoma. Clin Cancer Res. ’ was cited as reference 83.
  9. ‘Prins RM, Cloughesy TF, Liau LM. Cytomegalovirus immunity after vaccination with autologous glioblastoma lysate. N Engl J Med. 2008 Jul 31;359(5):539-41. doi: 10.1056/NEJMc0804818. PMID: 18669440; PMCID: PMC2775501.’ was not selected because the article presents CMV reactivation after autologous tumor cell lysate DC vaccine, rather than targeting CMV therapy.
  10. ‘Ahmed N, Brawley V, Hegde M, et al. HER2-Specific Chimeric Antigen Receptor-Modified Virus-Specific T Cells for Progressive Glioblastoma: A Phase 1 Dose-Escalation Trial. JAMA Oncol. 2017 Aug 1;3(8):1094-1101. doi: 10.1001/jamaoncol.2017.0184. PMID: 28426845; PMCID: PMC5747970.’ was not selected due to no relevance to CMV topic.

Ongoing clinical trials with no published articles were searched on website clinicaltrials.gov until 1st July 2022.

Round 2

Reviewer 2 Report

The revised manuscript is a clear review that readers understand the therapeutic potential of anti-CMV treatmment for GBM compared to the initial manuscript. References are properly cited and well-structured as a review article. 

This manuscript is a resubmission of an earlier submission. The following is a list of the peer review reports and author responses from that submission.